# Quantum interference between dark-excitons and zone-edged acoustic phonons in few-layer WS$_2$

Qing-Hai Tan [1,2,3,9], Yun-Mei Li[4,9], Jia-Min Lai[1,2], Yu-Jia Sun[1,2], Zhe Zhang[1,2], Feilong Song[1,2], Cedric Robert [5], Xavier Marie [5], Weibo Gao [3,6,7], Ping-Heng Tan [1,2] ✉ & Jun Zhang [1,2,8] ✉

Fano resonance which describes a quantum interference between continuum and discrete states, provides a unique method for studying strongly interacting physics. Here, we report a Fano resonance between dark excitons and zone-edged acoustic phonons in few-layer WS$_2$ by using the resonant Raman technique. The discrete phonons with large momentum at the M-point of the Brillouin zone and the continuum dark exciton states related to the optically forbidden transition at K and Q valleys are coupled by the exciton-phonon interactions. We observe rich Fano resonance behaviors across layers and modes defined by an asymmetry-parameter $q$: including constructive interference with two mirrored asymmetry Fano peaks (weak coupling, $q > 1$ and $q < -1$), and destructive interference with Fano dip (strong coupling, $|q| << 1$). Our results provide new insight into the exciton-phonon quantum interference in two-dimensional semiconductors, where such interferences play a key role in their transport, optical, and thermodynamic properties.

Resonance quantum interference is a general phenomenon, which strongly affects the electronic transport, optical, and vibronic properties of materials[1–4]. As one of the most representative phenomena of quantum interferences, Fano resonance describes interference between continuum states and discrete states, making it an ideal platform for studying the strongly interacting physics[1,2,5], such as the magnetization and electronic polarization[6–8], resonant electromagnetic effects[9], and exciton-phonon interactions (EPIs)[10–16]. In particular, Fano resonance Raman scattering induced by EPIs provides a powerful tool to reveal underlying physics in solid materials[6,12,13].

Recently, layered transition metal dichalcogenides (TMDs) and their heterostructures have received much attention due to their novel properties[17–19]. In these TMDs semiconductors, electronic energy band splitting induced by the spin-orbit coupling and multiple valleys (energy extrema) at different positions of the Brillouin zone form an abundance of exciton states[17], which provides an ideal playground for studying the resonance quantum interferences between excitons, photons and other quasiparticles such as phonons. Moreover, these valley features associated with the band splitting support optically-forbidden excitons, i.e., spin- and momentum-forbidden excitons, the so-called dark excitons[20–22], which play essential roles in their optoelectronic properties. The spin-forbidden dark excitons can be observed by measuring the photoluminescence spectrum with a giant in-plane magnetic field[23–27], in-plane detection[28]. The momentum-forbidden dark excitons can be detected by time-resolved exciton diffusion[29], or by directly imaging it in the momentum space by time-

[1]State Key Laboratory of Superlattices and Microstructures, Institute of Semiconductors, Chinese Academy of Sciences, Beijing 100083, China. [2]Center of Materials Science and Optoelectronics Engineering, University of Chinese Academy of Sciences, Beijing 100049, China. [3]Division of Physics and Applied Physics, School of Physical and Mathematical Sciences, Nanyang Technological University, 637371 Singapore, Singapore. [4]Department of Physics, Xiamen University, Xiamen 361005, China. [5]University of Toulouse, INSA-CNRS-UPS, LPCNO, 135 Av. Rangueil, 31077 Toulouse, France. [6]The Photonics Institute and Centre for Disruptive Photonic Technologies, Nanyang Technological University, 637371 Singapore, Singapore. [7]Centre for Quantum Technologies, National University of Singapore, Singapore 117543, Singapore. [8]CAS Center of Excellence in Topological Quantum Computation, University of Chinese Academy of Sciences, Beijing 100049, China. [9]These authors contributed equally: Qing-Hai Tan, Yun-Mei Li. ✉e-mail: phtan@semi.ac.cn; zhangjwill@semi.ac.cn

resolved angle-resolved photoemission spectroscopy (ARPES)[30]. These experiments show the dominant role of phonon-exciton scattering[29,30]. However, the response of phonons under such dark exciton-phonon interactions in these semiconductors remains largely unexplored. Moreover, the quantum interference between acoustic phonons and excitons in few-layer TMDs shows significant effects on their electrical transport properties[31] and their optical properties, such as the exciton/valley dynamics[32–36]. Therefore, it is important to study the dark excitons-acoustic phonons interactions/interferences in few-layer TMDs semiconductors.

In this work, we experimentally observed the dark exciton in bilayer WS$_2$ and studied the quantum interference between zone-edged acoustic phonon modes (the acoustic branches extending to the M-point, i.e., out-of-plane acoustic (ZA(M)), transverse acoustic (TA(M)), and longitudinal acoustic (LA(M)) modes) and dark excitons in a few-layer WS$_2$ semiconductor. We found that the coupling strength of these zone-edged acoustic phonons varies from weak coupling (constructive interference with a Fano peak) to strong coupling (destructive interference with a Fano dip) across layers and modes. We further revealed the symmetry roles on the quantum interference processes between dark excitons and phonons.

## Results

### Quantum interference and dark excitons

Figure 1 a shows the schematic diagram of the quantum interference between a discrete state and a continuum state, resulting in an asymmetry profile. This asymmetry profile can be described by the coupling term between these two states[5,6,37]:

$$I = I_0 \frac{(1 + \varepsilon/q)^2}{1 + \varepsilon^2}, \varepsilon = \frac{\omega - \omega_0}{\Gamma},$$
$$\hbar\Gamma = \pi V^2 \rho(E) + \hbar\gamma,$$
$$q = \left[ V \frac{T_P}{T_E} + V^2 R(E) \right] / \pi V^2 \rho(E),$$

(1)

where $I_0$, $\gamma$, and $\omega_0$ are the intensity, the linewidth, and the frequency of the uncoupled discrete state respectively. $\Gamma$ is the linewidth parameter. $V$ is the matrix element for the interaction between discrete and continuum states, and $\rho(E)$ is the density of continuum states. $T_p$ and $T_e$ are the scattering amplitude of the electronic continuum and decoupled discrete state, $R(E)/\pi$ term is the Hilbert transform of $\rho(E)$. $q$ is an asymmetry parameter, which gives the coupling strength and electronic information in varied materials[7,11,38]. The positive and negative $q$ correspond to different phase shifts (or relative energy shifts) of the two states, resulting in two mirrored asymmetry lineshape (as shown in Fig. 1a). Generally, $|q| = 1$ means dispersion with comparable phonon and electron contribution; $|q| \gg 1$ and $|q| \ll 1$ demonstrate a constructive and destructive dominated quantum interference process between discrete and continuum states[6], corresponding to the weak and strong coupling, respectively. Thus, a symmetric Lorentzian peak represents a negligible coupling case ($|q| \propto \infty$), while a Fano dip represents a strong coupling case ($|q| \ll 1$). Figure 1b shows a schematic diagram of the bright A exciton ($X_0$), the dark exciton ($D_1$), and the dark Q valley excitons ($D_2$) transition. We should note that the K and Q valleys of few-layer WS$_2$ are close to each other in energy[39]. Figure 1c shows the reflection spectra of 1-3L WS$_2$ at 4K and Fig. 1d shows the PL spectrum of a hexagonal boron nitride (hBN) encapsulated bilayer WS$_2$ excited/collected by two objectives with different numerical apertures (NA = 0.81 and 0.35) respectively. The high NA objective enables the detection of dark exciton ($D_1$) transition even at the normal incidence[28], as shown in Fig. 1d. We find that the dark state (-2.02 eV) lies below the bright state around $37 \pm 2$ meV, consistent with previously reported results in the monolayer case[23,27,28].

The long lifetime of dark exciton both at K and Q valley[30,40], promotes the accumulation of these Bose particles to form a bounded quasi-continuum state upon photoexcitation when considering the time scale of exciton-phonon interactions[41,42]. Therefore, this is a favorable situation to evidence the quantum interference between these dark exciton continuum states and phonons. Considering the momentum conservation, the transition of dark Q valley excitons needs the assistance of phonons. Meanwhile, under the non-resonant excitation case, only the phonon modes at the Γ point can be detected due to the same reason. The phonon scattering process mediated by defects or bounded exciton states can overcome such a momentum mismatch, making it possible to observe the first-order phonon modes at non-Γ point by Raman scattering[14,43–45]. We note that the wave vector of ΓM is

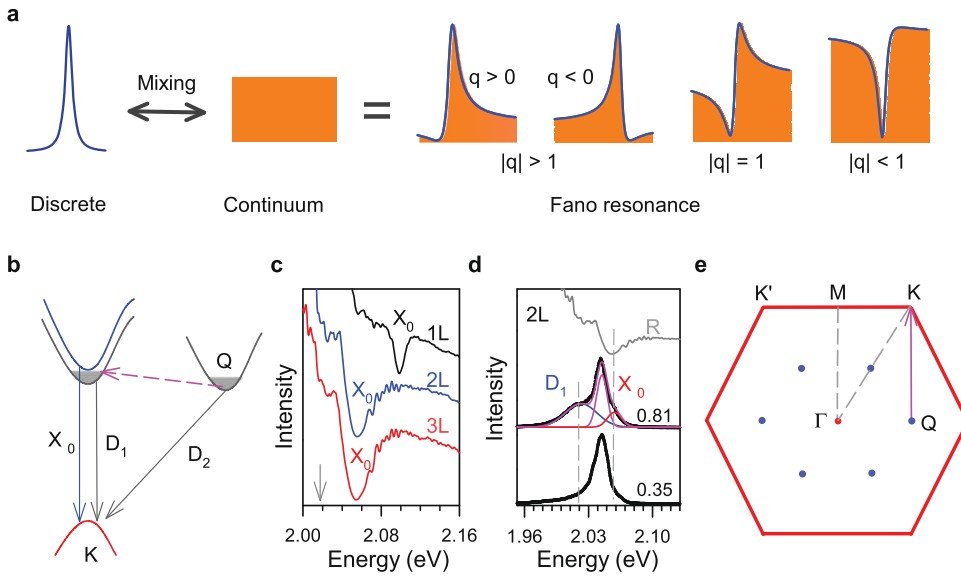

**Fig. 1 | Fano-typed quantum interference and dark excitons. a** Schematic diagram of Fano interferences between a discrete state and a continuum state. **b** A transition schematic diagram of the bright exciton ($X_0$), spin-forbidden dark exciton ($D_1$), and momentum-forbidden dark exciton ($D_2$). **c** Reflection spectra of 1-3L WS$_2$ (S1) at 4 K. **d** PL spectra of hexagonal boron nitride encapsulated bilayer WS$_2$ (S2) under a high numerical aperture (NA = 0.81) and a low numerical aperture (NA = 0.35) objective respectively. The bright exciton energy position is normalized to the exciton absorption energy of S1 (the gray line) to eliminate the small difference between two samples. The data are offset for clarity. **e** The first Brillouin zone of few-layer WS$_2$. The wave vector of ΓM is equal to QK.

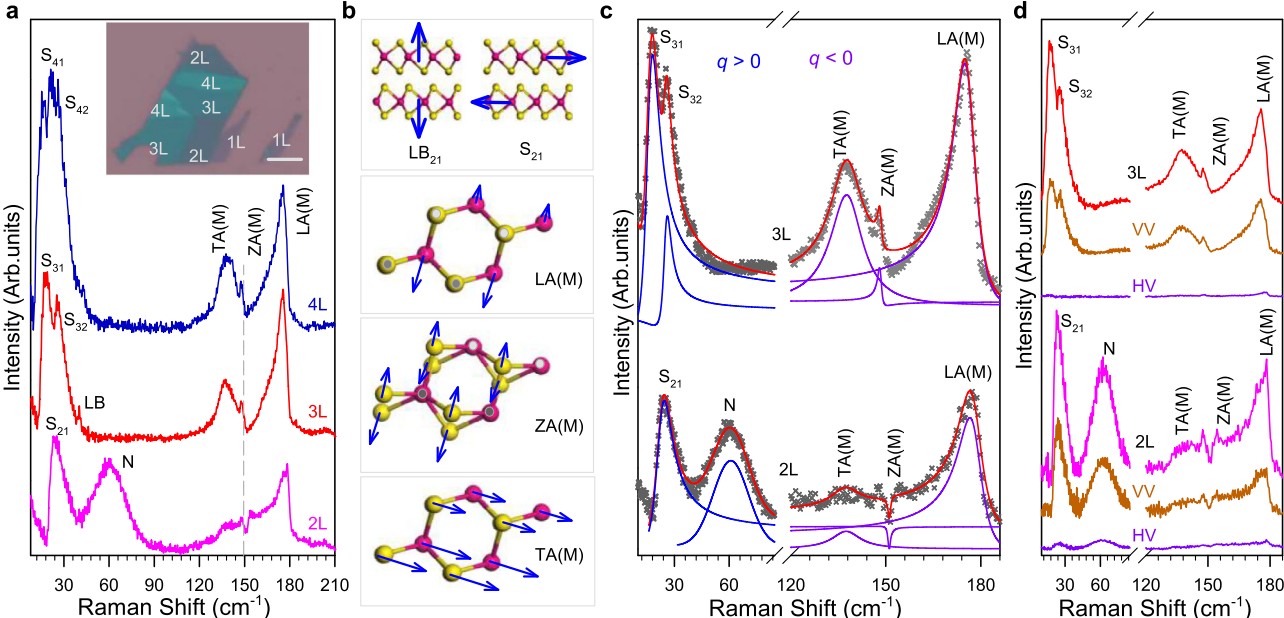

**Fig. 2 | Observation of quantum interference in few-layer WS$_2$. a** Raman spectra of 2-4L WS$_2$ at 4 K. Shear and layer breathing modes are labeled as S$_{mi}$ and LB$_{mi}$, where $m$ and $i$ represent layer numbers and the $i$th modes, respectively[46]. N denotes a new peak. The inset shows the optical microscopy image of the few-layer WS$_2$ sample (S1). The scale bar is 10 µm. **b** Illustration of the vibration of the layer breathing (LB$_{21}$), shear (S$_{21}$) modes of 2L-WS$_2$ and TA(M), ZA(M), and LA(M) modes (see more detailed schematics in SI). **c** The fitting results of shear and zone-edged acoustic modes in 2L- and 3L-WS$_2$. (**d**) Polarized Raman spectra of 2L- and 3L-WS$_2$ at 4 K. The excitation wavelength is 612 nm (2.026 eV). V (H) denotes the vertical (horizontal) polarization. VV and HV indicate parallel and cross-polarization configuration, respectively. The data are offset for clarity.

equal to QK in the first Brillouin zone of WS$_2$, see Fig. 1e. It suggests that the phonon modes at the M point can be directly observed with the mediation of scattering K valley excitons to Q valley. Considering the splitting of the valence band, a similar picture applies to B excitons as well, see Supplementary Fig. S1 in Supplementary Information (SI).

## Observation of quantum interference in WS$_2$

The few-layer WS$_2$ samples are exfoliated mechanically from their bulk crystals onto SiO$_2$/Si substrate, as shown in the inset in Fig. 2a. Their number of layers is determined by using the optical contrast and ultralow frequency Raman spectroscopy[46], as shown in Supplementary Fig. S2. The measurements presented in Fig. 1d have been obtained with a WS$_2$ bilayer encapsulated in hBN, which is prepared by using a standard dry transfer method. This yields a significant reduction of the inhomogeneous broadening[47]. As a consequence, the dark exciton D$_1$ transition lying close to the bright one can be evidenced. Because the photoluminescence background signal of monolayer WS$_2$ is too strong to obtain an available Raman signal when the excitation energies are close to the A exciton states, we focus our attention on bilayer (2L) to quadra-layer (4L) WS$_2$. Figure 2a shows the low-frequency (10 cm$^{-1}$ to 210 cm$^{-1}$) Raman spectra of 2-4L WS$_2$ with 612 nm (2.026 eV) excitation at 4 K. Figure 2b shows the vibration way of shear and layer breathing modes of 2L-WS$_2$, and TA(M), ZA(M), and LA(M) modes (see more detailed vibration schematic of these modes in SI). Figure 2c shows the fittings of the major peaks of 2-3L WS$_2$. We note that the excitation energy (2.026 eV) is close to the dark A exciton (D$_1$). Under this condition, the shear modes in 2-4L WS$_2$ are greatly enhanced and featured with an asymmetry Fano profile, whereas the layer breathing modes are weak and featured with a Lorentzian profile. The observed Fano profile of shear modes is due to the quantum interference between shear phonons and dark A excitons[14,16].

Besides these optical phonon modes at Γ point, three zone-edged acoustic phonon modes at M point, i.e., TA(M) ($B_{1g}$ symmetry), ZA(M) ($B_{3g}$ symmetry), and LA(M) ($A_g$ symmetry) modes[48], are also observed in 2-4L WS$_2$. Remarkably, ZA(M) and LA(M) modes also show a Fano

profile, which is mirrored with that for shear modes. In particular, the Fano coupling strength of ZA(M) and LA(M) modes varies across layers and modes, which is different from that for shear modes. Specifically, the ZA(M) mode in the 2L case behaves as a Fano dip ($|q| < 1$), unlike a Fano peak for other modes and layers (see Fig. 2a).This Fano dip can be understood by considering the electromagnetically induced transparency (EIT), which is a result of destructive Fano interference among different transition pathways[49–51] and occurs at $\omega_{light}$ - $\omega_{phonon}$ = $\omega_{exciton}$. Specifically, in our case, the valence band together with the conduction band in the K valley and Q valley form an equivalent three-level system here, eliminating the absorption due to the quantum interference between ZA(M) phonon and continuum dark states. As a result, when near the condition $\omega_{light} - \omega_{phonon} = \omega_{exciton}$, the destructive interference gives a narrow transparency window, i.e., the Fano dip. In 3L(4L)-WS$_2$, the condition above is broken may be attributed to the slight Q valley shift induced by the additional layers[52,53] and thus, giving peaks (instead of dips).

As discussed above, the observed three zone-edged acoustic phonons: TA(M), ZA(M), and LA(M) modes, can be explained by considering the bounded dark Q excitons mediated phonon scattering process (Fig. 1b, e). These bounded dark Q valley excitons (D$_2$) form a quasi-continuum state, similarly to the dark excitons (D$_1$) in the K valley. As a result, the quantum interference between dark K/Q excitons continuum and zone-edged acoustic phonon modes leads to the observation of the Fano profile (see more detailed discussions and schematic of phonon-involved scattering process in Supplementary Fig. S3). The mirrored Fano profiles between shear phonon at Γ point ($q > 0$) and zone-edged acoustic modes at M point ($q < 0$), suggest the different phase shift (or relative energy shift) of two interference states. Specifically, the shear phonons (at Γ point) are mainly involved in intravalley dark exciton scattering processes, whereas the zone-edged acoustic phonons are mainly involved in intervalley exciton scattering processes (higher level K excitons and lower level Q excitons), as shown in Fig. S3. Meanwhile, the phase of phonons at Γ point changes by $\pi$ relative to that at M point also can lead to this mirrored feature.

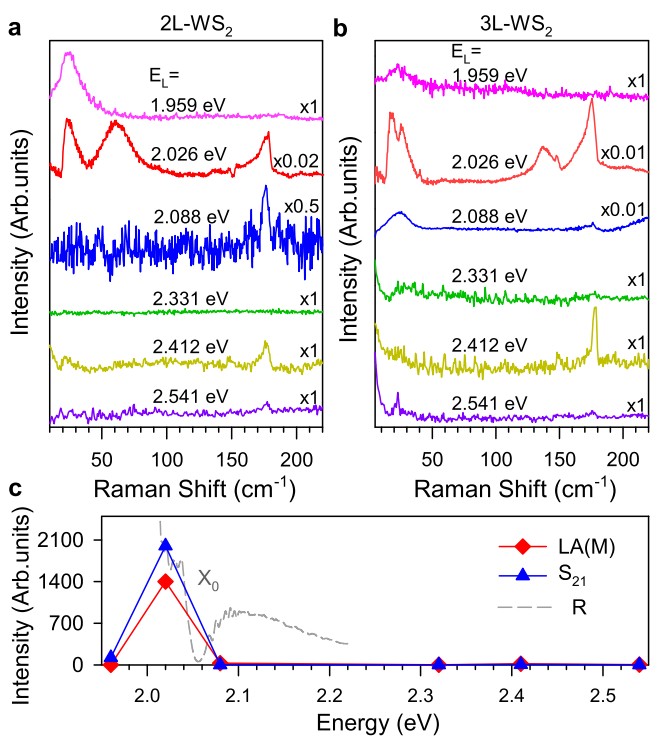

**Fig. 3 | Excitation energy dependence of quantum interference. a, b** Raman spectra of 2L- and 3L WS$_2$ under different excitation energies (E$_L$) at 4 K, respectively. The intensities are normalized to the Raman modes of silicon substrate (524 cm$^{-1}$ at 4 K). The data are offset for clarity. **c** The intensity of shear and LA(M) modes respect to the excitation energies. The dotted gray line is the reflection spectrum of 2L-WS$_2$.

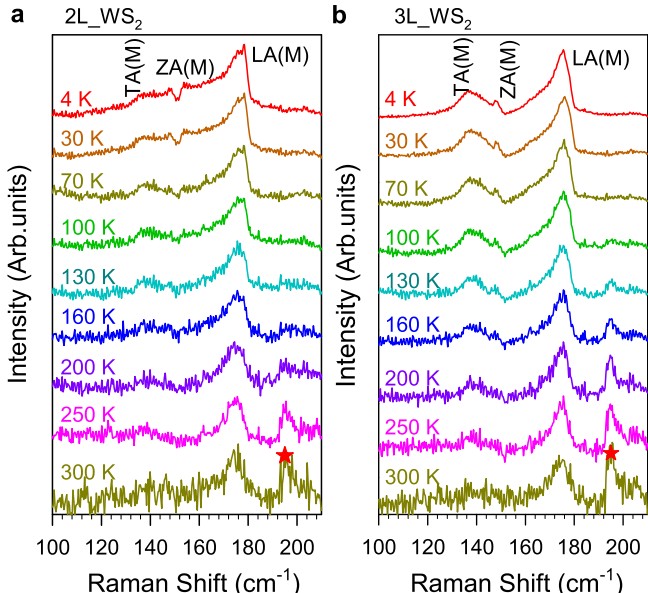

**Fig. 4 | Temperature dependence of quantum interference. a, b** Temperature-dependent Raman spectra of TA(M) and ZA(M) and LA(M) phonon modes in 2L and 3L WS$_2$ with 612 nm excitation. The intensities are normalized to the LA(M) mode. The star symbol indicates the second-order phonon mode: E′(M) - LA(M). The data are offset for clarity.

Additionally, we find a new peak (N) at ~60 cm$^{-1}$ only in 2L WS$_2$ at 4 K. This N mode can be observed in all 2L WS$_2$ regions, and thus, the defect-induced phonon modes and localized excitons can be excluded. To further confirm the origin of this peak, more studies are required in the future. Figure 2d shows the polarized-Raman spectra of 2-3L WS$_2$ at 4 K. We find that all Raman modes almost vanish under cross-polarization configuration (HV) (although these in-plane shear modes should survive under the normal case, see more discussion in SI). This result can be understood due to the breakdown of Raman selection rules by the Fröhlich interaction between dark excitons and phonons[15].

### Excitation energy dependence of quantum interference

When the excitation energies are away from the D$_1$, the intensity of zone-edged acoustic phonons is much weaker than that with 612 nm excitation, as shown in Fig. 3 and Supplementary Figs. S4–S7. For example, when the excitation energy (2.086 eV, 594 nm) is slightly higher than the energy of X$_0$, the Fano resonance vanishes. Figure 3c shows the resonant profile of shear and LA(M) mode of 2L WS$_2$. We find that the resonant peak is close to the dark exciton D$_1$, different from that for A$_{1g}$ (see Fig. S4 in SI). These results can be explained by considering that, when the excitation energies are higher than X$_0$, i.e., above optical bandgap excitation, instead of scattering K excitons to Q valley through zone-edged acoustic phonons, there are more relaxation channels for excitons/electrons to Q valley without the assistance of zone-edged acoustic phonon[30]. Similarly, when the laser energy (1.96 eV, 633 nm) is slightly below dark exciton D$_1$, the Fano resonance also disappears due to the off-resonance. For these excitation energy far away from the exciton energies, e.g., 2.331 eV, the intensity of these shear and zone-edged acoustic modes is weak, as shown in Fig. 3. These results prove that the continuum states in the quantum interference are indeed from dark excitons. Meanwhile, the power-dependent measurements suggest these processes correspond to first-order Raman scattering, as shown in Supplementary Fig. S8 (see more discussions in SI).

### Temperature dependence of quantum interference

Figure 4 shows the temperature-dependent Raman spectra of 2-3L WS$_2$ with 612 nm excitation. The Fano dip of ZA(M) mode in 2L WS$_2$ nearly vanishes at around 100 Kelvin (K), and the Fano peak of ZA(M) and LA(M) modes almost disappeared at around 160 K. These results can be well explained by considering that the WS$_2$ exciton energy decreases when the temperature increases (see Supplementary Fig. S9). Therefore, at higher temperatures, the off-resonance with dark excitons will destroy such Fano resonance processes. On the other side, although X$_0$ is getting close to the excitation energy (612 nm) up to 160 K (see Supplementary Fig. S9), the Fano resonance still becomes weaker, unlike other modes, e.g., E′(M)-LA(M). The above results further prove that the Fano resonance results from the interference between acoustic phonons and dark excitons.

### Discussions

Generally, the Fano-type quantum interference will appear when the continuum states and discrete states have the same symmetry (e.g., propagation/polarization direction) and meet the energy matching[10]. Therefore, the symmetry-determined coupling between zone-edged acoustic (or shear) phonons and dark excitons is the key factor in understanding why only these modes appeared with the Fano profile under the resonant excitation[20]. First, we note that the shear and layer breathing phonons are the relative vibrations between layers, different from those between atoms within a single layer, i.e., high-frequency optical phonons. If we only consider the vibration way of a single layer, the shear and layer breathing modes can be treated as "quasi-acoustic" phonons. It implies that the quantum interference occurs mainly between dark excitons and acoustic phonons.

The EPI strength is given by[2]

$$H_{ep} = \frac{1}{\sqrt{N}} \sum_{\mathbf{k},\mathbf{q},mn\nu} g_{mn\nu}(\mathbf{k},\mathbf{q}) \hat{a}^{\dagger}_{m,\mathbf{k}+\mathbf{q}} \hat{a}_{n,\mathbf{k}} (\hat{b}_{\mathbf{q}\nu} + \hat{b}^{\dagger}_{-\mathbf{q}\nu}), \qquad (2)$$

the coupling matrix $g_{mn\nu}(\mathbf{k},\mathbf{q}) = \langle u_{m,\mathbf{k}+\mathbf{q}}|\Delta_{\mathbf{q}\nu}|u_{n,\mathbf{k}}\rangle_{uc}$, where $u_{n,\mathbf{k}}$ is the periodic part of Bloch wavefunction and the integral is over the whole unit cell. $\hat{a}_{n,\mathbf{k}}$ ($\hat{a}^{\dagger}_{n,\mathbf{k}}$) is the electron annihilation (creation)

operator, while $\hat{b}_{\mathbf{q}\nu}(\hat{b}_{\mathbf{q}\nu}^{\dagger})$ denotes the phonon's with momentum $q$, mode $\nu$ and frequency $\omega_{\mathbf{q}\nu}$. The strength of EPI is determined by how strong the lattice vibration affects the wavefunction of the excitons[2,54]. Here the wavefunctions of the conduction and valence bands near the K point are mainly contributed from $d$ orbitals of the W atoms, which are greatly confined within a single layer. In particular, the propagation (polarization) of dark A exciton ($\Gamma_4$ symmetry) is along the in-plane (Z) direction[24,25,28], a strong interference with the in-plane vibrational modes is expected. By contrast, the variation of out-of-plane vibration shows a slight effect on the dark exciton at K point, and thus, the coupling between them is weak, see more discussions in SI. Consequently, the shear phonons, instead of layer breathing phonons, show Fano profiles. For Fano resonance of zone-edged acoustic phonons, both dark excitons in K and Q valley are involved in this interference process. Since the electronic states in Q valley are made from both W and S atoms orbitals, the strong interference/coupling occurring only in-plane is removed.

Finally, we analyze the coupling strength difference for three zone-edged acoustic phonons and dark excitons. It can be understood by considering the effective coupling between their different vibration ways (see Supplementary Figs. S10, S11) and momentum direction. Specifically, when neglecting the "umklapp" (folding) processes, only the first-order term of the coupling matrix $g_{mn\nu}(\mathbf{k}, \mathbf{q})$ needs to be considered[2]:

$$g_{mn\nu}(\mathbf{k},\mathbf{q}) \propto g_0 \mathbf{q} \cdot \mathbf{e}_{\mathbf{q}\nu}, \tag{3}$$

where $\mathbf{e}_{\mathbf{q}\nu}$ is the polarization of the acoustic wave with wave vector $\mathbf{q}$, and mode $\nu$ and $g_0$ are the parameters depending on the materials. Considering the symmetry of these zone-edged acoustic phonon modes, the electron-phonon coupling strength of longitudinal modes is stronger than that for the transverse mode ($\mathbf{q} \cdot \mathbf{e}_{\mathbf{q}\nu}$) in WS$_2$. Consequently, the TA mode holds a weaker interference effect and appears with a nearly symmetric profile in spectra.

In summary, we found that the dark state is ~37 meV below the bright state in bilayer WS$_2$. We further revealed that the first-order zone-edged acoustic phonon modes in few-layer WS$_2$ can be directly observed thanks to the momentum match between ΓM (for acoustic phonons) and QK (for scattering K excitons to Q valley). The quantum interference strength between dark excitons and these phonons varies from constructive to destructive across layers and modes, which is strongly determined by the vibration way of phonons and the symmetry of dark excitons. Since the electronic energy band structure of few-layer WSe$_2$ and MoS$_2$ is similar to that of few-layer WS$_2$[23,26,39], such quantum interferences are expected to be observed with proper excitation energies. Furthermore, the twisted bilayer or multilayer TMDs-based heterostructures and homostructures will also provide an additional platform to study and tune such quantum interferences between excitons and phonons, as well as the interference effects on their physical properties. Our results can give deep insight into the dark excitons-phonons interferences in layered semiconductors and pave the way for designing novel devices based on such excitons-phonons quantum interferences.

## Methods
### Sample preparation
The few-layer WS$_2$ samples are exfoliated mechanically from their bulk crystals onto SiO$_2$/Si substrate. For WS$_2$ bilayer encapsulated in hBN, which is prepared by using a standard dry transfer method.

### Optical measurements
Raman measurements were undertaken in backscattering geometry with a Jobin-Yvon HR800 system equipped with a liquid-nitrogen-cooled charge-coupled detector. The spectra were collected with a 50 × long-working-distance objective lens (NA = 0.5) at low temperature measurements. The ultralow-frequency Raman spectra were obtained down to ±5 cm$^{-1}$ by combining three volume Bragg grating filters into the Raman system to efficiently suppress the Rayleigh signal. The Montana cryostat system was employed to cool the samples down to 4 K under a vacuum of 0.1 mTorr. The reflectance contrast $\Delta R/R$ were undertaken with a 50 × objective lens (NA = 0.5) and an 300 lines mm$^{-1}$ grating with white light sources. The PL spectra were undertaken with objective lens with different numerical apertures (NA = 0.81 and 0.35).

## Data availability
The data that support the findings of this study are available from the corresponding authors on reasonable request.

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

## Acknowledgements

We thank Maciej Molas for helpful discussions. J. Z. acknowledges support of the National Key Research and Development Program of China (Grant No. 2017YFA0303401), Beijing Natural Science Foundation (Grant No. JQ18014), National Natural Science Foundation of China (Grant No. 12074371), CAS Interdisciplinary Innovation Team, Strategic Priority Research Program of Chinese Academy of Sciences (grant NO. XDB28000000). P. H. Tan acknowledges support of the National Natural Science Foundation of China (Grant nos. 11874350) and CAS Key Research Program of Frontier Sciences (Grant no. ZDBS-LY-SLH004 and XDPB22). W. B. Gao thanks the support of the Singapore NRF through its CRP Program (CRP Award Nos. NRF-CRP21-2018-0007, NRF-CRP22-2019-0004).

## Author contributions

J.Z. and P.T. supervised the project; J.Z., P.T., and Q.T. conceived the ideas; Q.T. and J.L. prepared the samples. P.T. designed Raman experiments; Q.T., J.L., Y.S., Z.Z., and F.L. performed experiments; Q.T., Y.L., C.R., X.M., W.G., P.T. and Z.J. analyzed the data; Q.T., Y.L. and Z.J. wrote the manuscript with input from all authors.

## Competing interests

The authors declare no competing interests.
