## [Peer Review File · Nature Communications]

Reviewers' Comments:

Reviewer #1:

Remarks to the Author:

In this work, the authors present a detailed study of the Raman spectra of few-layer WS₂ when the excitation energy is in resonance with the dark exciton. The authors analyze the line shape of the phonons in terms of the Fano resonance between the continuum of the dark exciton state and the discrete excitation of zone-boundary acoustic phonons. The experimental data are new and interesting. A Fano-like line shape of the low-frequency shear modes in WS₂ has been reported before [FlatChem 3, 64]. However, the phonon lines analyzed in this work are different from previous study. There are a few issues that need to be cleared before this work can be considered for publication.

1. First of all, I am not 100% convinced that the observed effects are due to a Fano resonance. Usually, the Fano resonance is a resonance between a continuum state and a discrete state when the energy of the discrete state overlaps with the continuum. In the current case, the continuum is the exciton state and the discrete excitation is the zone-boundary phonons. Although the Raman signal of the phonons overlap with the exciton energy, the phonons themselves have much smaller energy. Therefore, the phonon cannot be in resonance with the exciton continuum. Rather, the scattered photon from the Raman scattering event is in resonance with the exciton state. I am not sure if the mechanism of the Fano resonance can be applied to this case.
2. The Raman spectra of interlayer vibration modes at low temperature (Fig. 2 & 3) and those measured at room temperature are very much different. The most striking is the absence of the interlayer modes in the Raman spectra measured at 4 K with non-resonant excitation. Also, the shear modes are broader at low temperature. How can this difference be explained?
3. In Fig. 3, the Shear mode and the LA(M) mode show up for the 2.412 eV excitation whereas they are suppressed for 2.331 or 2.541 eV excitations. Can this be explained?
4. The authors interpret the dip in the spectrum of 2L as being due to electromagnetically induced transparency. I am not sure if EIT and the Fano resonance are directly correlated. If so, a proper reference and/or theoretical explanation is warranted. A related question is why this 'dip' occurs only for the 2L case.

Reviewer #2:

Remarks to the Author:

This is a well-prepared manuscript with interesting findings. Fano-type features appear in resonant Raman spectra of few-layer WS₂. These features are explained by interference between dark excitons and Raman modes, including the zone-center layer shearing mode and zone-edge modes. The evidence well supports the explanation.

Can the authors comment on why there seems to be no signal for 2.33 eV in Fig. 3a?

Can the authors include a comment on whether similar features might potentially appear for other TMDs as well?

The manuscript should be carefully double-checked for missing words in some sentences and ensure all abbreviations have been defined before being used.

I recommend publication after these minor revisions.

Reviewer #3:

Remarks to the Author:

This manuscript describes a Fano resonance between dark excitons and zone-edge acoustic phonons in 2-4 layer WS₂ measured with resonant Raman spectroscopy. Aside from minor grammar concerns, the paper is well structured and easy to read. The results are convincing, and the claims are well supported. However, I am not convinced that this manuscript offers sufficient

significance/novelty to merit publication in Nature Communications. Reference 14 is a manuscript published by these authors in 2D Materials in 2017 that also described resonant Raman spectroscopy of an exciton-phonon Fano resonance in WS₂. While the results in this submission nicely complement the results in the 2017 2D Materials manuscript, I suspect they might be more appropriate for a more specialized journal.

If the authors feel that they can make a compelling case for the impact and novelty of these results compared with reference 14, I have only one additional minor concern that I would like to see addressed. While there is an extensive literature describing these sorts of Fano resonances as quantum interference, Fano resonances are frequently observed in purely classical systems as well. It is not necessarily clear to me that the effect observed here are properly described as a quantum interference effect (nor is it clear that this is anything more than a linguistic quibble; the results are interesting whether or not they can be described as a classical ensemble interference effect). Nonetheless, I'd like to see more concrete support for the idea that this isn't a classical Fano resonance effect if claims of quantum interference are to appear so prominently.

Reviewer #1:

In this work, the authors present a detailed study of the Raman spectra of few-layer WS₂ when the excitation energy is in resonance with the dark exciton. The authors analyze the line shape of the phonons in terms of the Fano resonance between the continuum of the dark exciton state and the discrete excitation of zone-boundary acoustic phonons. The experimental data are new and interesting. A Fano-like line shape of the low-frequency shear modes in WS₂ has been reported before [FlatChem 3, 64]. However, the phonon lines analyzed in this work are different from previous study. There are a few issues that need to be cleared before this work can be considered for publication.

Reply: We sincerely thank the reviewer for carefully reading our manuscript and giving positive comments. We also thank the reviewer for bringing to our attention the relevant article, of which we were not aware. We have cited this paper in the revised manuscript. Below we give a point-to-point reply to the comments.

1. First of all, I am not 100% convinced that the observed effects are due to a Fano resonance. Usually, the Fano resonance is a resonance between a continuum state and a discrete state when the energy of the discrete state overlaps with the continuum. In the current case, the continuum is the exciton state and the discrete excitation is the zone-boundary phonons. Although the Raman signal of the phonons overlap with the exciton energy, the phonons themselves have much smaller energy. Therefore, the phonon cannot be in resonance with the exciton continuum. Rather, the scattered photon from the Raman scattering event is in resonance with the exciton state. I am not sure if the mechanism of the Fano resonance can be applied to this case.

Reply: We thank the reviewer for raising this point. The Fano resonance mechanism in our manuscript can be understood by considering the physical picture described below.

For Stokes Raman scattering, the relationship between incident light (E_{in}), scattering light (E_s), and phonon (E_{ph}) is $E_{in} = E_s + E_{ph}$. Meanwhile, for simplicity, the relationship between exciton-accumulated dark level (E_{ED}), pure dark level (E_D), and continuum bandwidth formed by dark excitons (E_{con}) is $E_{ED} = E_D + E_{con}$. Here the formation of the continuum states is a result of the accumulation of dark exciton both at K and Q valley thanks to their long lifetime feature when considering the time scale of exciton-phonon interactions and density of states in the 2D system, as discussed in the main text. We should note that there are no continuum states below the E_D . For resonant Raman scattering in our case, the energy of incident light (E_{in}) (scattering light (E_s)) is in resonance with the exciton state E_{ED} (E_D) instead of E_{ph} , as the reviewer commented. The phonons' energies are ~ 3 meV and ~ 20 meV for shear phonons and zone-edged acoustic phonons respectively, which are close to the magnitude of the

continuum width (~ 40 meV). As a result, here the Fano resonance arises from discrete phonons and the continuum (E_{con}) state of the dark exciton, as shown in Fig. R1.

The discussions above have been added to the Supplementary Information in the revised manuscript.

Fig. R1 | **a, b**, Schematics of dark exciton-phonon resonance-induced quantum interference processes, including Γ point phonon-involved intravalley scattering (a) and M point phonon-involved intervalley scattering (b).

2. The Raman spectra of interlayer vibration modes at low temperature (Fig. 2 & 3) and those measured at room temperature are very much different. The most striking is the absence of the interlayer modes in the Raman spectra measured at 4 K with non-resonant excitation. Also, the shear modes are broader at low temperature. How can this difference be explained?

Reply: We thank the reviewer for the comment. At low temperatures, the phonon scattering effects on the exciton are greatly reduced. As a result, the reduced phonon-exciton interactions under non-resonant excitation lead to the weak Raman signal of phonon modes and the narrowing of exciton linewidth¹. As shown in Fig. R2, the interlayer shear phonon mode S_{31} still can be resolved with a weak intensity under 2.33 eV non-resonant excitation, suggesting that a longer acquisition time and higher excitation power can give a clearer signal of these interlayer phonon modes. On the other hand, the resonant condition (exciton energy) at low temperature and room temperature (see Fig. S8) is changed, leading to the difference in Raman spectra at room- and low temperatures under the same excitation condition.

The linewidth broadening of shear modes is due to the Fano resonance. Specifically, the linewidth in the coupled system is $\Gamma = \Gamma_0 \left| \frac{q^2+1}{q^2-1} \right|$, where Γ_0 is the uncoupled linewidth parameter and q is an asymmetry parameter. Whereas for non-Fano resonant cases, e.g., 2.54 eV excitation, the linewidth of shear mode is slightly reduced at low temperatures (see Fig. 3 and Fig. S2).

Fig. R2 | a, The log-scale Raman spectrum of 3L-WS2 with 2.331 eV excitation.

3. In Fig. 3, the Shear mode and the LA(M) mode show up for the 2.412 eV excitation whereas they are suppressed for 2.331 or 2.541 eV excitations. Can this be explained?

Reply: We thank the reviewer for the comment. The shear mode and the LA(M) mode show up for 2.412 eV excitation because this excitation energy is closer to the bright B exciton energy (~ 2.43 eV), which is different from the non-resonant excitations of 2.331 eV and 2.54 eV, as shown in Fig. R3b (also Fig. S1b).

The valance band splitting results in the B exciton, as shown in Fig. R3a (also Fig. S1). In principle, the scattering process described in the main text for A exciton can be applied to B exciton also. Therefore, these acoustic modes are expected to be observed when the excitation energies are close to the B exciton. However, we noted that these two acoustic modes, as well as the shear modes at Γ point, are weak when excitation energies are close to the B exciton. These results can be understood that 2.412 eV excitation belongs to the above bandgap excitation, and thus, there are more relaxation channels for scattering electrons/excitons to Q valley without the assistance of acoustic phonon at M point². Meanwhile, the dark B exciton state is the upper band one (2.47 eV), so the forming of a continuum state by exciton population is more difficult than that for dark A exciton. As a result, under such a resonance condition, the Fano resonance of shear modes basically vanishes, and the Fano resonance of acoustic modes at the M point is greatly weakened. These measured results under B exciton also further support the Fano resonance from the interference between phonons and dark A/Q excitons.

Fig. R3 | a, Schematic of valence band splitting induced A and B excitons. b, Reflectance contrast spectrum of 1L-WS₂ at 4 K. A red arrow indicates the energy of 2.33 eV.

4. The authors interpret the dip in the spectrum of 2L as being due to electromagnetically induced transparency. I am not sure if EIT and the Fano resonance are directly correlated. If so, a proper reference and/or theoretical explanation is warranted. A related question is why this ‘dip’ occurs only for the 2L case.

Reply: We thank the reviewer for the suggestions and comments. The electromagnetically induced transparency (EIT) is indeed a result of Fano interferences among different transition pathways³⁻⁵. Specifically, the destructive interference leads to this EIT. Most of the previous studies on EIT are focused on atomic systems. Whereas as for the observation of EIT in transition metal dichalcogenides via Raman spectroscopy, as far as we know, we are the first to report it.

The observed Fano dip only in 2L WS₂ may be due to the layer effects on the band structures. When compared to the K band, Q valley bands are more sensitive to interlayer coupling. It means that additional layers, i.e., 3L or 4L cases, will slightly change the Q valley energy, and thus, the destructive interference condition is broken.

In the revised manuscript, we have added relevant discussions to make them much clearer and cited the above references.

References

1. Moody, G. et al. Intrinsic homogeneous linewidth and broadening mechanisms of excitons in monolayer transition metal dichalcogenides. *Nature Communications* 6, 8315 (2015).
2. Madéo, J. et al. Directly visualizing the momentum-forbidden dark excitons and their dynamics in atomically thin semiconductors. *Science* 370, 1199-1204 (2020).
3. Fleischhauer, M., Imamoglu, A. & Marangos, J. P. Electromagnetically induced transparency: Optics in coherent media. *Reviews of Modern Physics* 77, 633-673 (2005).

4. Peng, B., Özdemir, Ş. K., Chen, W., Nori, F. & Yang, L. What is and what is not electromagnetically induced transparency in whispering-gallery microcavities. *Nature Communications* 5, 5082 (2014).
5. Limonov, M., Rybin, M., Poddubny, A. et al., Fano resonances in photonics. *Nature Photonics* 11, 543–554 (2017).

Reviewer #2:

This is a well-prepared manuscript with interesting findings. Fano-type features appear in resonant Raman spectra of few-layer WS₂. These features are explained by interference between dark excitons and Raman modes, including the zone-center layer shearing mode and zone-edge modes. The evidence well supports the explanation.

Reply: We sincerely thank the reviewer for the positive comments. Below we address the referee’s comments in detail.

1. Can the authors comment on why there seems to be no signal for 2.33 eV in Fig. 3a?

Reply: We thank the reviewer for the comment. For 2.33 eV excitation, the Raman signal is weak when compared to those under other excitation energies, see Fig.3 and Fig. R1(c). Because this excitation energy is far away from the resonance conditions, leading to the weak signal, as shown in Fig. R1. Specifically, the bright (dark) A exciton energy of few-layer WS₂ is ~2.06 eV (~2.02 eV), whereas the bright (dark) B exciton is ~2.43 eV (~2.47 eV). We note that all of them are away from the energy of 2.33 eV. As a result, the Raman signal under 2.33 eV excitation at low temperatures is weak.

Fig. R1 | a, The log-scale Raman spectrum of 3L-WS₂ with 2.331 eV excitation. **b**, Reflectance contrast spectrum of 1L-WS₂ at 4 K. A red arrow indicates the 2.33 eV.

2. Can the authors include a comment on whether similar features might potentially appear for other TMDs as well?

Reply: We thank the reviewer for the constructive comment. The electronic energy band structure of few-layer WSe₂ and MoS₂ is similar to that of few-layer WS₂, i.e., the spin-forbidden dark state is slightly below the bright state, and the momentum forbidden dark state at Q valley is close to the K valley as well¹⁻⁴. Hence, when the excitation

energies are close to the dark A exciton of WSe₂ or MoS₂, such Fano resonances of zone-edged acoustic phonons are expected to be observed. Furthermore, similar features are also expected to appear in the van der Waals heterostructures or homostructures that are formed by stacking these transition metal dichalcogenides together.

The discussions above have been added to the revised manuscript.

3. The manuscript should be carefully double-checked for missing words in some sentences and ensure all abbreviations have been defined before being used.

Reply: We thank the reviewer for the suggestion. We have carefully checked the language and abbreviations and corrected them accordingly.

4. I recommend publication after these minor revisions.

Reply: We thank the reviewer for the recommendation.

References

1. Mak, K. F., Lee, C., Hone, J., Shan, J. & Heinz, T. F. Atomically Thin MoS₂: A New Direct-Gap Semiconductor. *Physical Review Letters* 105, 136805 (2010).
2. Splendiani, A. et al. Emerging Photoluminescence in Monolayer MoS₂. *Nano Letters* 10, 1271-1275 (2010).
3. Zhang, X.-X. et al. Magnetic brightening and control of dark excitons in monolayer WSe₂. *Nature Nanotechnology* 12, 883-888 (2017).
4. Robert, C. et al. Measurement of the spin-forbidden dark excitons in MoS₂ and MoSe₂ monolayers. *Nature Communications* 11, 4037 (2020).

Reviewer #3:

This manuscript describes a Fano resonance between dark excitons and zone-edge acoustic phonons in 2-4 layer WS₂ measured with resonant Raman spectroscopy. Aside from minor grammar concerns, the paper is well structured and easy to read. The results are convincing, and the claims are well supported.

Reply: We thank the reviewer for the positive assessment of our work. Below we address the reviewer's comments in detail.

1. However, I am not convinced that this manuscript offers sufficient significance/novelty to merit publication in *Nature Communications*. Reference 14 is a manuscript published by these authors in *2D Materials* in 2017 that also described resonant Raman spectroscopy of an exciton-phonon Fano resonance in WS₂. While the results in this submission nicely complement the results in the 2017 *2D Materials* manuscript, I suspect they might be more appropriate for a more specialized journal.

Reply: We thank the reviewer for the comments. The main research object, physical mechanism, and analysis of this work are different from the previous study, as commented by Reviewer #1. We would like to list the major new findings/physics of this manuscript and provide some background/discussion to show the significance/novelty of our work below.

The main new findings/physics

1) In this work, we mainly studied the Fano resonance of zone-edge-acoustic phonons at M point, which is different from the Brillouin zone Γ point phonons in previous studies. As it is well known, for first-order Raman scattering, usually only phonon modes at Γ point, i.e., the optical phonon modes, can be detected due to the momentum conservation. The defects or bounded states can provide momentum compensation during the scattering process, resulting in the observation of non- Γ point first-order phonon modes. Therefore, the physical picture of the Γ point and non- Γ point scattering processes is quite different. The most famous example is the explanation of D mode in graphene^{1,2}. Hereby considering the momentum match between Γ -M (for acoustic phonons at M point) and Q-K (for scattering K excitons to Q valley), we observed the zone-edge acoustic phonons at M point. As far as we know, our work is the first to report the Fano resonances of zone-edge acoustic phonons at M point due to the momentum-forbidden dark excitons.

2) In contrast to the Fano resonance of shear modes, in which the coupling strength is almost constant for different layers and modes, the Fano resonance of ZA(M) and LA(M) modes can vary from a constructive interference to a destructive interference due to the layer's effects on the Q valley band^{3,4}, suggesting the coupling strength ranges from weak coupling to strong coupling across layers and modes. For example, both the Fano dip (destructive interference) of ZA(M) and the Fano peak (constructive interference) of LA(M) are observed in 2L-WS₂. We believe that these Raman results are also reported for the first time in few-layer TMDs materials.

3) In this work, we first reported the PL spectrum of dark exciton in bilayer WS₂ and revealed that the symmetry-determined coupling between dark excitons and phonons is the key factor for these quantum interference processes.

The significance/novelty

Layered transition metal dichalcogenides (TMDs) have attracted much attention thanks to their novel properties. Most of their physical properties are dominated by excitons and phonons (especially acoustic phonons). Naturally, the exciton-phonon coupling also strongly affects their transport, optical, and thermodynamic properties. In particular, in these TMDs semiconductors, the dark excitons dominate the exciton state distribution in these materials and most of their optical properties⁵⁻⁷. However, there are still limited methods that can be used to study these dark states, not to mention the dark exciton-acoustic phonon coupling, which also strongly influences their optical and

electrical transport properties⁸⁻¹⁰. Among these dark excitons, the momentum-forbidden dark excitons, which consist of an electron and a hole residing at different valleys, play a dominant role in few-layer TMDs due to the indirect bandgap. For these intervalley electron/exciton scatterings, the Brillouin-zone-corner (valley) phonons must be involved. However, exactly which zone-corner phonons are involved in these processes remains elusive. And it is challenging to study these first-order zone-corner phonons directly. By considering the momentum match between Γ -M (for acoustic phonons at M point) and Q-K (for scattering K excitons to Q), our work provides a direct way to study the zone-edged phonons (at M point), as well as the coupling between these phonons and excitons.

Fano resonance describes quantum interference between continuum states and discrete states, which is always important to know the underlying physics in condensed matter physics. It has been successfully used to study the magnetization and electronic polarization phenomena in semiconductors and superconductors, as well as the electron-phonon coupling in resonant Raman scattering. Moreover, the Fano resonance effect can be used to create the ultra-coherent Fano nanolaser^{11,12} and to control the nonlinear optical processes, e.g., surface-enhanced Raman scattering. Therefore, understanding and application of such a Fano resonance process in TMDs are significant both for fundamental physics and potential applications.

In summary, our work can provide quantitative new insight into the properties of dark excitons, zone-edged acoustic phonons, and their interference/interaction in few-layer WS₂. We believe that similar quantum interferences are expected to appear in other TMDs and conventional semiconductors with similar band structures.

2. If the authors feel that they can make a compelling case for the impact and novelty of these results compared with reference 14, I have only one additional minor concern that I would like to see addressed. While there is an extensive literature describing these sorts of Fano resonances as quantum interference, Fano resonances are frequently observed in purely classical systems as well. It is not necessarily clear to me that the effect observed here are properly described as a quantum interference effect (nor is it clear that this is anything more than a linguistic quibble; the results are interesting whether or not they can be described as a classical ensemble interference effect). Nonetheless, I'd like to see more concrete support for the idea that this isn't a classical Fano resonance effect if claims of quantum interference are to appear so prominently.

Reply: We thank the reviewer for raising this point. The Fano resonances concept was first proposed to atomic physics by Ugo Fano to understand the asymmetry profiles in absorption based on a superposition principle from quantum mechanics¹³. The underlying physical mechanism is the quantum interference between a continuum state and a single autoionizing resonance state (a discrete excited state of an atom)^{13,14}. Up

to now, besides atomic systems, Fano-typed quantum interference has also been observed in semiconductor quantum dots¹⁵, electrical transport¹⁶, and classical ensemble systems^{17,18}, et al, revealing its universality in various systems in terms of interference-induced asymmetry profiles. However, we should note that classical systems usually discuss the interference effect of classical electromagnetic wave or elastic wave scattering in the media and do not involve the interactions with the (quasi)particles¹⁷⁻¹⁹. We can make a classical analog of the quantized (quasi)particle for many quantum systems to get a classical ensemble. Whereas the treatment of the interactions between different (quasi)particles is usually beyond the classical interpretation.

In our case, the particles or quasiparticles involved in the Fano resonance, i.e., the photon, phonon, and exciton, are all quantized. Although the phonon near the Γ point can also be described by the long-wavelength acoustic waves, the phonons at the M point can't be characterized by the classical waves. Moreover, the interactions between these (quasi)particles are quantum effects, for example, the creation or annihilation of phonons during the scattering process must be described by the quantum field theory. In fact, such quantum interference effects between excitons and acoustic phonons on the exciton transport have been discussed in theory⁹. The quantum interference has also been experimentally reported in Raman scattering in layered ReS_2 ²⁰, doped semiconductors^{21,22}, and graphene^{23,24}. In our case, the Fano resonance relates to the zone-edge acoustic phonon vibration coupled to the continuum of dark exciton states through exciton-phonon interactions. Such coupling can lead to new elementary excitations described by hybrid acoustic phonon-dark exciton states, in which the zone-edge acoustic phonons are 'dressed' with the dark exciton cloud and strongly renormalized by their interactions, including its centre frequency, linewidth, and Raman activity. Therefore, we are confident that the Fano resonance here can be described as a quantum interference effect.

References

- 1 Ferrari, A. C. Raman spectroscopy of graphene and graphite: Disorder, electron-phonon coupling, doping and nonadiabatic effects. *Solid State Communications* **143**, 47-57 (2007).
- 2 Ferrari, A. C. *et al.* Raman Spectrum of Graphene and Graphene Layers. *Physical Review Letters* **97**, 187401 (2006).
- 3 Mak, K. F., Lee, C., Hone, J., Shan, J. & Heinz, T. F. Atomically Thin MoS_2 : A New Direct-Gap Semiconductor. *Physical Review Letters* **105**, 136805 (2010).
- 4 Splendiani, A. *et al.* Emerging Photoluminescence in Monolayer MoS_2 . *Nano Letters* **10**, 1271-1275 (2010).

- 5 Madéo, J. *et al.* Directly visualizing the momentum-forbidden dark excitons and their dynamics in atomically thin semiconductors. *Science* **370**, 1199-1204 (2020).
- 6 Zhang, X.-X. *et al.* Magnetic brightening and control of dark excitons in monolayer WSe₂. *Nature Nanotechnology* **12**, 883-888 (2017).
- 7 Wang, G. *et al.* In-Plane Propagation of Light in Transition Metal Dichalcogenide Monolayers: Optical Selection Rules. *Physical Review Letters* **119**, 047401 (2017).
- 8 Helmrich, S. *et al.* Phonon-Assisted Intervalley Scattering Determines Ultrafast Exciton Dynamics in MoSe₂ Bilayers. *Physical Review Letters* **127**, 157403 (2021).
- 9 Glazov, M. M. Quantum Interference Effect on Exciton Transport in Monolayer Semiconductors. *Physical Review Letters* **124**, 166802 (2020).
- 10 Jiang, C. *et al.* Microsecond dark-exciton valley polarization memory in two-dimensional heterostructures. *Nature Communications* **9**, 753 (2018).
- 11 Mork, J., Chen, Y. & Heuck, M. Photonic Crystal Fano Laser: Terahertz Modulation and Ultrashort Pulse Generation. *Physical Review Letters* **113**, 163901 (2014).
- 12 Yu, Y. *et al.* Ultra-coherent Fano laser based on a bound state in the continuum. *Nature Photonics* **15**, 758-764 (2021).
- 13 Fano, U. Effects of Configuration Interaction on Intensities and Phase Shifts. *Physical Review* **124**, 1866-1878 (1961).
- 14 Miroshnichenko, A. E., Flach, S. & Kivshar, Y. S. Fano resonances in nanoscale structures. *Reviews of Modern Physics* **82**, 2257-2298 (2010).
- 15 Kroner, M. *et al.* The nonlinear Fano effect. *Nature* **451**, 311-314 (2008).
- 16 Göres, J. *et al.* Fano resonances in electronic transport through a single-electron transistor. *Physical Review B* **62**, 2188-2194 (2000).
- 17 Luk'yanchuk, B. *et al.* The Fano resonance in plasmonic nanostructures and metamaterials. *Nature Materials* **9**, 707-715 (2010).
- 18 Rybin, M. V. *et al.* Fano resonances in antennas: General control over radiation patterns. *Physical Review B* **88**, 205106 (2013).
- 19 Kato, H. & Kato, H. Fano Resonance of an Elastic Waveguide with an Island Structure. *Journal of the Physical Society of Japan* **89**, 024402 (2020).
- 20 Zhang, S. *et al.* Quantum interference directed chiral raman scattering in two-dimensional enantiomers. *Nature Communications* **13**, 1254 (2022).
- 21 Cerdeira, F., Fjeldly, T. A. & Cardona, M. Interaction between electronic and vibronic Raman scattering in heavily doped silicon. *Solid State Communications* **13**, 325-328 (1973).
- 22 Chandrasekhar, M., Renucci, J. B. & Cardona, M. Effects of interband

excitations on Raman phonons in heavily doped n-Si. *Physical Review B* **17**, 1623-1633 (1978).

- 23 Tang, T.-T. *et al.* A tunable phonon–exciton Fano system in bilayer graphene. *Nature Nanotechnology* **5**, 32-36 (2010).
- 24 Tan, P. H. *et al.* The shear mode of multilayer graphene. *Nature Materials* **11**, 294-300 (2012).

Reviewers' Comments:

Reviewer #1:

Remarks to the Author:

The authors have answered the questions raised by the reviewers and revised the manuscript. I am satisfied with the answers to my question. I have no further comments.

Reviewer #2:

Remarks to the Author:

The authors have addressed all the reviewers' comments in great detail. I recommend publication in the present state.

Reviewer #3:

Remarks to the Author:

I'm satisfied with the authors' responses. However, I don't think any substantial error analysis is included. The editorial policy checklist notes that clearly defined error bars are present, but I don't see error bars anywhere in the manuscript. Some form of error analysis would probably be worthwhile here.

RESPONSE TO REVIEWER COMMENTS

We thank three reviewers for their constructive comments and recommendations, which help us to improve the manuscript. We would like to address all the comments point by point as below.

Reviewer #1 (Remarks to the Author):

The authors have answered the questions raised by the reviewers and revised the manuscript. I am satisfied with the answers to my question. I have no further comments.

Reply: We thank the reviewer for the recommendation.

Reviewer #2 (Remarks to the Author):

The authors have addressed all the reviewers' comments in great detail. I recommend publication in the present state.

Reply: We thank the reviewer for the recommendation.

Reviewer #3 (Remarks to the Author):

I'm satisfied with the authors' responses. However, I don't think any substantial error analysis is included. The editorial policy checklist notes that clearly defined error bars are present, but I don't see error bars anywhere in the manuscript. Some form of error analysis would probably be worthwhile here.

Reply: We thank the reviewer for the nice suggestion. In the revised version, we have added the fitting error for the bright-dark exciton splitting in the main text and the error bars for the temperature-dependent bright exciton energy in Fig. S9 in the Supplementary Information. As for the fitting errors of parameter q in Fano profile fitting, we have already included them in Table 1 in the Supplementary Information.